# Patient preference and acceptability of self-sampling for cervical screening in colposcopy clinic attenders: A cross-sectional semi-structured survey

Sophie Webb[1,2]*, Nafeesa Mat Ali[1], Amy Sawyer[2], David J. Clark[1], Megan A. Brown[1], Yolanda Augustin[1,2], Yin Ling Woo[3,4], Su Pei Khoo[3,4], Sally Hargreaves[1], Henry M. Staines[1], Sanjeev Krishna[1,2,3,5,6], Kevin Hayes[1,2,7]*

1 Clinical Academic Group in Institute for Infection & Immunity, St George's University of London, London, United Kingdom, 2 St George's University Hospitals NHS Foundation Trust, London, United Kingdom, 3 Faculty of Medicine, University of Malaya, Kuala Lumpur, Malaysia, 4 ROSE Foundation, Kuala Lumpur, Malaysia, 5 Institut für Tropenmedizin, Universitätsklinikum Tübingen, Tübingen, Germany, 6 Centre de Recherches Médicales de Lambaréné, Lambaréné, Gabon, 7 Institute for Medical and Biomedical Education, St George's University of London, London, United Kingdom

* m2109631@sgul.ac.uk (SW); khayes@sgul.ac.uk (KH)

**Data Availability Statement:** Available on St. George's University of London data repository: https://doi.org/10.24376/rd.sgul.24716424.v1.

## Abstract

Low vaginal self-sampling has been pioneered as an important development to improve uptake of cervical screening globally. Limited research is available in specific patient groups in the UK exploring views around self-sampling to detect high-risk human papillomavirus (hrHPV) DNA. Therefore, we explored patient views to support development of a novel point-of-care self-sampling cervical cancer screening device, by undertaking a cross-sectional semi-structured questionnaire survey to explore preferences, acceptability, barriers and facilitators around self-sampling. Patients attending a colposcopy clinic, 25–64 years old, were invited to participate after having carried out a low vaginal self-sample using a regular flocked swab. Participants self-completed an anonymous 12-point questionnaire. Quantitative data were analysed in MS Excel and Graphpad Prism, and qualitative data with Nvivo. We recruited 274 patients with a questionnaire response rate of 76%. Acceptability of self-sampling was high (95%, n = 187/197; Cronbachs-α = 0.778). Participants were asked their choice of future screening method: a) low vaginal self-sampling, b) healthcare professional collected vaginal swab, c) cervical brush sample with healthcare professional speculum examination, or d) no preference. Preferences were: a) 37% (n = 74/198), b) 19% (n = 37/198); c) 9% (n = 17/198), and d) 35% (n = 70/198), showing no single option as a strong preference. Key motivators were: Test simplicity (90%, n = 170/190), speed (81%, n = 153/190) and less pain (65%, n = 123/190). Barriers included lack of confidence taking the sample (53%, n = 10/19), resulting in preference for a healthcare professional sample (47%, n = 9/19). Whilst self-sampling showed high acceptability, lack of strong preference for screening method may reflect that respondents attending colposcopy are already engaged with screening and have differing perception of cervical cancer risk. This group appear less likely to 'switch' to self-sampling, and it may be better targeted within primary and community

**Funding:** This work was support by the St. George's Hospital Charity. HMS is supported by the Wellcome Trust Institutional Strategic Support Fund (204809/Z/16/Z) awarded to St George's University of London. www.wellcome.org SH is funded by the National Institute for Health and Care Research (NIHR300072 and NIHR134801) www.nihr.ac.uk, the Academy of Medical Sciences - www.acmedsci.ac.uk (SBF005\1111), the La Caixa Foundation (La Caixa LCF/PR/SP21/52930003) - https://lacaixafoundation.org, Research England - www.ukri.org/councils/research-england, and WHO - www.who.int. The funders had no role in study design, data collection and analysis, decision to publish, or preparation of the manuscript.

**Competing interests:** I have read the journal's policy and the authors of this manuscript have the following competing interests: SK and HMS are advisors to and shareholders in QuantuMDx, a molecular nucleic acid test-based diagnostic company. SK, YA and HMS are advisors to Global Access Diagnostics, a developer of rapid diagnostic tests, and SK is a member of the Scientific Advisory Committee for the Foundation for Innovative New Diagnostics (FIND), a not-for-profit organisation that produces global guidance on affordable diagnostics. These competing interests will not alter adherence to PLOS Global Public Health policies on sharing data and materials.

care, focusing on under-screened populations. Any shift in this paradigm in the UK requires comprehensive education and support for patients and providers.

## Introduction

Cervical cancer is the fourth most common cancer in women globally [1]. The majority (99.8%) of cases are caused by persistent high-risk human papillomavirus (hrHPV) infection and are preventable through a combination of HPV vaccination and screening for hrHPV. The World Health Organization (WHO) has called on global stakeholders to work towards cervical cancer elimination worldwide by ensuring 90% of female adolescents are vaccinated, 70% of eligible women are screened with a high-performance HPV test at least twice in a lifetime and ensuring 90% of women who are screened positive receive appropriate follow-up and treatment by 2030 [2]. Limited access to screening, barriers to treatment, and lack of access to HPV vaccination in developing countries [2] are amongst the vital issues that need to be tackled to achieve the WHO's ambition on cervical cancer elimination.

Cervical screening coverage in the United Kingdom continues to decline, with only 68.7% of eligible women screened in 2022–23 (down by 1.2% from the previous year) [3]. This figure was further impacted by the COVID-19 pandemic, which saw screening invitations briefly suspended, and primary care providers given the option to postpone invitations, if needed, for up to 6 months [4]. Along with this, the National Health Service (NHS)–and in particular primary care providers–are facing a crisis, with mounting pressures from long waiting times, staff shortages, and the need to implement catch-up screening. As a result, many women are facing difficulties and delays accessing appointments for cervical screening [5].

Low awareness and understanding of screening are also factors in the declining screening rates, with 1 in 5 women in the UK unaware that cervical screening does not detect ovarian cancer [6, 7]. Inequalities in access to screening remain a challenge, with 63% of physically disabled women reporting being unable to attend screening [8], and 80% of women in full-time employment unable to find a convenient appointment [7]. Furthermore, women in areas with higher levels of deprivation [9], and survivors of sexual violence [10] are also less likely to attend screening. A 2023 report by Jo's Cervical Cancer Trust, which surveyed 848 individuals working within cervical cancer and prevention in the UK [7] further highlighted these challenges, and found that the main barriers to screening were: Anxieties among the eligible population (66%), workforce pressures in primary care (63%), low levels of understanding among the public (62%), inaccessibility for some of the eligible population (48%) and lack of digitalisation (32%).

In response to the call for solutions to improve access, steps are being taken towards innovative alternative methods of screening, including low vaginal self-sampling for hrHPV. This advance could represent an important landmark in the UK screening programme's evolution, having transitioned from liquid-based cytology to hrHPV nucleic acid amplification tests (NAATs) primary screening in 2019. Low vaginal self-sampling for hrHPV DNA testing for cervical cancer screening has already been implemented nationally in a number of countries worldwide including the Netherlands [11], Malaysia [12] and Australia [13]. Self-sampling approaches have the aim of tackling well documented common barriers to screening such as embarrassment, pain, fear and preference for a female healthcare practitioner [14].

In a meta-analysis from 2018, self-sampling for hrHPV using PCR-based assays was found to be as accurate as healthcare professional acquired samples in detecting cervical

intraepithelial neoplasia grade 2 or worse (CIN2+) [15]. There have been concerns about reduced 'real-world' detection of CIN2+, compared to the meta-analysis, particularly in 'switchers' who currently regularly attend healthcare professional-based screening and may 'switch' to self-sampling. These 'switchers' will only be managed according to their hrHPV status, whereas in primary care they would have the additional information of the cytology result from the healthcare professional acquired cervical brush sample. The reasons for this are not yet fully understood, but it is hypothesized that regular screeners may have more recently acquired hrHPV infections and possibly lower viral load on screening. Another concern is a subsequent lack of adherence to the follow-up cytology sample that would require a healthcare professional-taken sample [16].

The United Kingdom Health Security Agency (UKHSA) is considering offering a choice of self-sampling or healthcare professional acquired sampling to all eligible individuals at the point of invitation [17]. It is important therefore, that evidence is gathered from multiple different populations, on the acceptability and preference for self-sampling in the UK. We therefore aimed to seek the views, acceptability and preferences of a group of women attending a colposcopy clinic at a teaching hospital in south west London that serves an ethnically diverse population (London boroughs of Merton—59% non-white British, and Wandsworth—52% non-white British ethnicity, according to 2021 UK Census data [18]). Participants were attendees at both screening and follow up colposcopy and therefore in the 'switchers' group of interest.

## Methods

### Design

This cross-sectional survey was designed along-side an on-going study to develop and evaluate a novel point-of-care cervical cancer screening device, to specifically look at acceptability of a self-sampling option for hrHPV detection. Participants who consented to participate in the study on attending their colposcopy appointment at St. George's University Hospitals NHS Foundation Trust, London were asked to complete a semi-structured, anonymous, self-completed questionnaire. Participant enrolment took place prospectively between 13th July 2022 and 19th January 2023.

### Ethics statement

Ethical approval for participation was provided by the Institutional Review Board as part of the "Development of a prototype device for cervical cancer screening" study, sponsored by St George's University of London (Integrated Research Application System project ID: 235626; Southwest—Cornwall & Plymouth Research Ethics Committee reference: 18/SW/0244.) All participants were adults and gave written informed consent to be part of the wider study and questionnaire survey.

Participants undertook a low vaginal self-sample using a Copan FLOQSwab 552c regular flocked swab, followed by a healthcare professional acquired low vaginal swab and an endocervical swab, before proceeding with their routine colposcopy appointment. Participants were a mix of new and follow-up patients, and therefore included both hrHPV positive and negative women. All participants were asked to complete the questionnaire after their appointment. In order to maintain anonymity, personal details and demographics were not recorded on the questionnaire. The questionnaire was estimated to take 10 minutes to self-complete, it was semi-structured and can be reviewed in full in S1 Questionnaire. The data was entered into RedCap for ease of analysis and quantitative data were analysed in MS Excel and Graphpad Prism and checked by two members of the study group, whilst qualitative data were analysed

with thematic analysis using Nvivo. For quantitative data, an internal reliability analysis was performed for consistency using a Cronbach-α coefficient.

## Participants

All attendees at the colposcopy clinic who were eligible at screening and consented to participate in the self-sampling study, between 25–64 years of age, were approached. Exclusion criteria were as follows: a history of cervical cancer; hysterectomy; pregnancy or post-partum less than 4 months; menstruation on the day of clinic; or use of douche/contraceptive cream or NuvaRing (etonogestrel/ethinyl estradiol vaginal ring)/HRT cream/vaginal pessary for prolapse/Replens (non-hormonal vaginal moisturiser)/thrush cream; severe illness warranting urgent care. On arrival at the clinic, patients were invited to participate in the study, given a patient information sheet (PIS) and allocated time to read and understand the PIS. A member of the study team then approached patients to ask if they had any further questions and if they were willing to participate, written informed consent was taken. Participation in the study had no impact on their clinical care, and they could opt out of the study at any time until they had handed in the questionnaire to the study team.

## Validation

The questionnaire used in this study was adapted from a previously validated questionnaire [19, 20] that had been used widely in Malaysia [12] to assess acceptability of low vaginal self-sampling for HPV testing in their local population [21, 22]. To ensure the questionnaire was suited to a UK clinical setting, we sought input from the public and patient involvement and engagement (PPIE) group at St George's University of London and consulted a group of healthcare professionals including clinical nurse specialists and gynaecology doctors. In response to feedback from members of the public with regards to patient understanding, we re-worded some of the questionnaire items for clarity, to make it more relevant to the UK population, and also added a free text section at the end to allow for additional comments that participants felt were pertinent. Unlike the Malaysian studies we did not take any pre-sampling demographics and the questionnaire was only performed following the self-sampling, as opposed to before and after, and our study population were specifically attending a colposcopy clinic for themselves, as opposed to a more opportunistic screening approach in the Malaysian study.

## Measures

The primary outcome was to assess acceptability of low vaginal self-sampling and future screening preference. Secondary outcomes were to assess the participant's experience of low vaginal self-sampling and the reasons for intention to utilise or decline options for low vaginal self-sampling in the future.

The questionnaire consisted of 12 questions (S1 Questionnaire), 7 of which used a 5-point Likert scale, one of which was to assess ease of understanding of the patient information sheet for the concurrent novel point-of-care device cervical cancer screening study. The other six questions were aimed indices of the patient experience of taking a low vaginal self-sample using a swab, with a score of 1 being the most negative response, and 5 the most positive. A score of 4 or more was considered as a positive response, implying acceptability of self-sampling. Participants were then asked about their most preferred method of cervical screening (from the following options: a) low vaginal self-sampling, b) a healthcare professional collected vaginal swab, c) a cervical brush sample or 'pap smear' with healthcare professional speculum

examination, or d) no preference.), and if they would be willing to perform a low vaginal self-sample again as part of future screening and follow up.

There was also a question on feedback for the participant information sheet for the cervical cancer screening study within which the questionnaire was embedded, and a final question with an open text box for any other feedback that the respondents felt to be relevant.

### Inclusivity in global research

Additional information regarding the ethical, cultural, and scientific considerations specific to inclusivity in global research is included in the Supporting Information (S1 Checklist).

## Results

A total of 274 women were initially recruited for the hrHPV POC validation study within which this questionnaire was embedded, of which two withdrew prior to collecting samples and were not therefore invited to complete the questionnaire (Fig 1). The final number of women who completed the questionnaire was 207, a response rate of 76% (207/272). All responses were included in analysis, including those with incompletely filled questionnaires.

### Future screening preference (primary outcome)

Two hundred and seven questionnaires were returned, of which 197 participants (95%) provided a response to the question "Would you be willing to do the HPV self-sampling test again", with the majority (95%, n = 187) reporting that they were willing to repeat low vaginal self-sampling in the future.

Having taken their own self-swab for the cervical cancer screening study, women selected their most preferred screening method from 4 choices: self low vaginal swab, healthcare professional -collected low vaginal swab, 'pap smear' with healthcare professional performing speculum, or no preference. A total of 198 women responded to this question, with 74 (37%) preferring low vaginal self-sampling, 70 (35%) reporting no preference, 37 (19%) preferring a

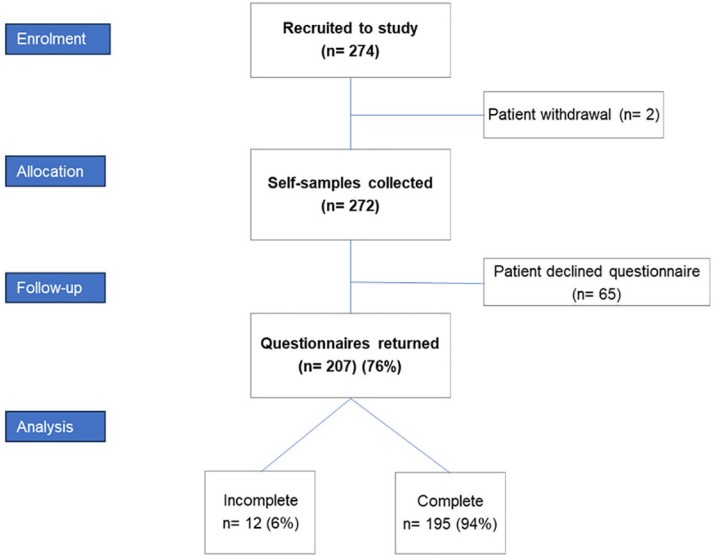

**Fig 1. Consort diagram for questionnaire response rate.**

**Table 1. Preferred method of cervical screening, after having attempted a low vaginal self-sample.**

| Which method do you prefer the MOST for cervical screening? | n (total = 198) | % |
|---|---|---|
| Self-sampling vaginal swab for HPV testing | 74 | 37% |
| Healthcare professional—collected vaginal swab for HPV testing | 37 | 19% |
| Pap smear—healthcare professional conducting a speculum examination | 17 | 9% |
| No preference | 70 | 35% |

healthcare professional-collected low vaginal swab and 17 (9%) preferring the traditional speculum examination healthcare professional-collected sample (Table 1).

## Reasons for screening preference

A multiple-choice question ('select all that apply') was asked after the preferred choice of screening, to ascertain motivators and barriers. The main reasons that motivated participants to perform low vaginal self-sampling again in the future were test simplicity (n = 170/190; 90%), speed (n = 153/190; 81%) and that the test was not painful (n = 123/190; 65%); whilst the main barriers were lack of confidence in taking the sample correctly (n = 10/19; 53%) and therefore preferring a clinician to take the sample (n = 9/19; 47%) (Table 2).

The free text response box was analysed for any other factors highlighted that were not included in the main questionnaire. Whilst these responses were predominantly messages of encouragement and praise for the research itself, a small number of additional factors were highlighted; need for more information on accuracy of the result from a self-sample (n = 7/196 [3%]), and access to screening appointments for healthcare professional taken samples was also noted as a barrier (n = 2/196; 1%).

## Patient acceptability of self-sampling

The first 6 questions of the questionnaire looked at the various elements of women's experience in taking a low vaginal self-sample, and all six questions were answered by 100% of respondents. An internal reliability analysis was performed which showed good consistency, with a Cronbach-$\alpha$ coefficient of 0.778. Table 3 shows the 1–5 Likert-scale rating responses, and Fig 2 shows a visual representation of these data.

**Table 2. Motivators and barriers to screening preference.**

| Motivators to hrHPV self-sampling | n | % of respondents (n = 190) |
|---|---|---|
| Simple | 170 | 90% |
| Quick | 153 | 81% |
| Not painful | 123 | 65% |
| Confident taking accurately | 80 | 42% |
| More comfortable | 63 | 33% |
| Less embarrassed | 58 | 31% |
| **Barriers to hrHPV self-sampling** | **n** | **% of respondents (n = 19)** |
| Not confident taking accurately | 10 | 53% |
| Prefer clinician to take | 9 | 47% |
| Not easy | 1 | 5% |
| Afraid will hurt self | 1 | 5% |
| Not comfortable | 1 | 5% |
| Painful | 0 | 0% |

**Table 3. Likert-scale responses of patient experience of self-sampling.**

|  | Overall experience | Ease | Convenience | Embarrassment | Discomfort/pain | Confidence |
|---|---|---|---|---|---|---|
| **V Positive (5)** | 136 | 163 | 182 | 172 | 139 | 52 |
| **Positive (4)** | 57 | 37 | 21 | 27 | 45 | 115 |
| **Neutral (3)** | 14 | 6 | 4 | 5 | 12 | 34 |
| **Negative (2)** | 0 | 1 | 0 | 3 | 11 | 5 |
| **V Negative (1)** | 0 | 0 | 0 | 0 | 0 | 1 |

Women reported their overall experience of self-sampling was positive 93% of the time, and none reported a score of 2 or less. Both ease (97%) and convenience (98%) of self-sampling were also highly rated.

Embarrassment, and pain or discomfort, are known to be common barriers to traditional healthcare professional acquired samples [14], and the majority of women reported that self-sampling was not embarrassing (96%) or painful (89%).

Another common concern, confidence in ability to take a self-sample correctly [14], had the widest spread of responses, and the highest number of 'neutral' responses (16%), but 81% still scored their confidence as positive. Overall, most (92%) women scored the six indices of acceptability of self-sampling with positive answers i.e. 4 or more, indicating a high level of acceptability of this method of screening.

### Understanding of the patient information sheet

A single 5-point Likert scale question about participant understanding of the cervical cancer screening study was included at the end of the questionnaire, which showed excellent understanding of the information with 93% (182/195) rating it 4 or greater.

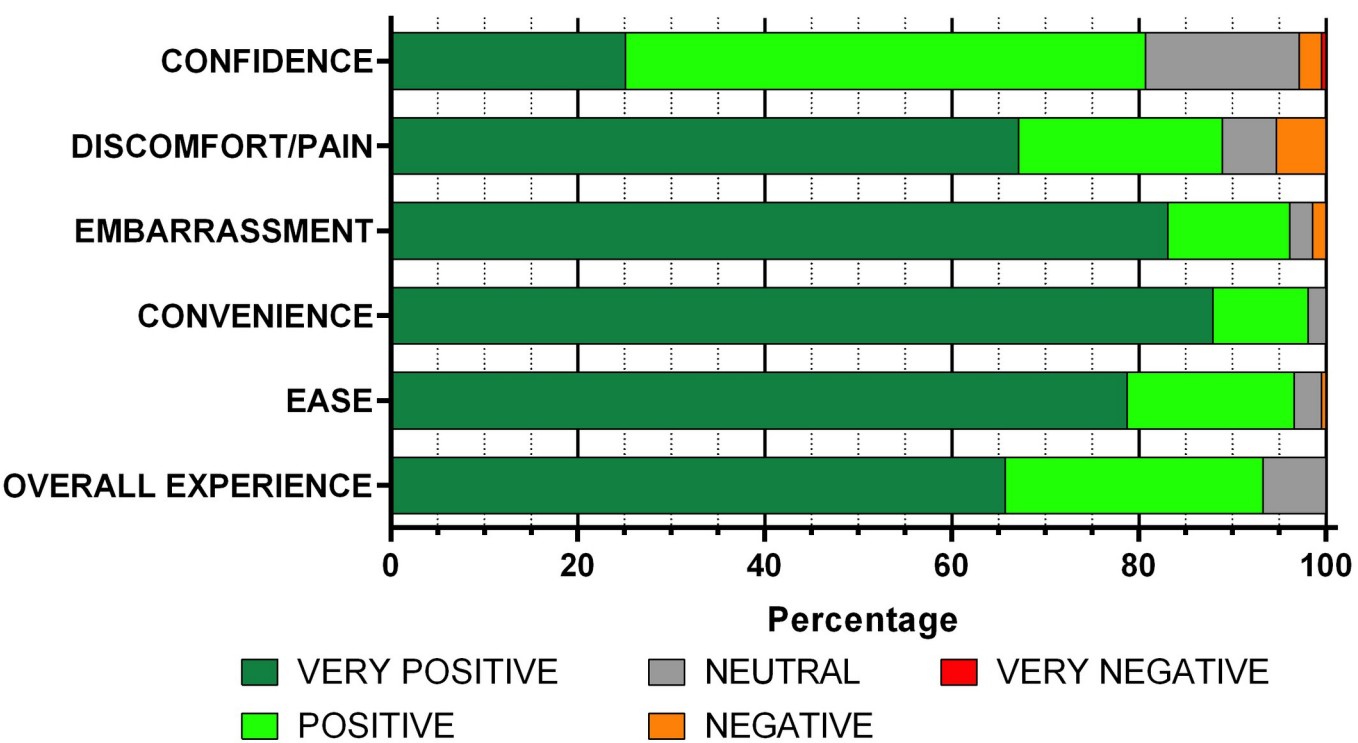

**Fig 2. 100% Stacked bar charts for Likert-scale responses of patient experience of self-sampling.**

## Discussion and future recommendations

This study examined the views of a group of patients who are likely to be regular attenders at screening, given their engagement with secondary care colposcopy services. As such, they are likely to be in the group of potential cervical screening 'switchers' [16] that UKHSA might consider when developing policy and deciding whether hrHPV self-sampling should be aimed at non-attenders or all screen-eligible individuals.

Interestingly, whilst participants indicated their willingness to perform a self-sample again (95%), their future screening preference was more indifferent, with just over one third preferring a self-sample, one third a healthcare professional-taken sample (either with or without speculum), and one third no preference. This is comparable to a recent UK study looking at preferences of all screen-eligible women at the point of invitation, which showed that whilst a higher proportion (59%) of women preferred self-sampling, around a third still felt more confident with a healthcare professional acquired sample [23]. The proportion of women who preferred self-sampling was also lower in our study compared to a recent systematic review [24] of acceptability of hrHPV self-sampling, but the review mostly included studies aimed at capturing screening non-attenders, and this key difference may help to explain the response in the cohort examined in this study. The perception of risk in the sampled population of highly engaged colposcopy clinic attenders may be different to non-attenders, or in women who attend screening but then have a negative hrHPV result, and so may have more trust and confidence in results from healthcare professional-taken samples [23].

The results presented here suggest that whilst around a third of regular screeners are likely to switch to self-sampling, and a third have no preference for self-sampling versus healthcare professional collected samples, a third prefer healthcare professional collected samples still, despite self-sampling being highly acceptable. The 'intention-behaviour gap' [25] may lower this proportion of 'switchers' further; and may alleviate fears of reduced detection of hrHPV in this group; whilst importantly allowing both regular screening attenders, and screening non-attenders a new choice for self-sampling and thus engagement with the cervical screening programme. It is also worth noting that, whilst not specifically investigated in this study, self-collected urine hrHPV testing is also emerging as a new strategy with good concordance to vaginal samples, which may extend the choices for screening even further, being less invasive and potentially easier to collect [26, 27].

In our study, self-sampling for hrHPV DNA was again shown to be highly acceptable (92%), in line with existing literature [24, 28], and the main reasons that women were motivated to perform self-sampling again were that it is simple (90%), quick (81%) and not painful (65%); similar to factors cited in comparable studies [24, 28]. Barriers were lack of confidence in taking the sample correctly (53%) and preferring a clinician to take the sample (47%), but it was also noted that the respondents were keen for further information on the accuracy of the self-sampling test compared to the pap smear and healthcare professional acquired samples; this important information may influence their decision making and is crucial to consider when rolling out any self-sampling programme in the UK. This is in line with previous findings by Jo's Cervical Cancer Trust [7] and bolsters the need for national awareness campaigns, increased education on screening and HPV, and indicates the need for further research on methods of teaching women to self-sample and the effect of this on confidence in screening method and choice of future screening method.

Further research also needs to be undertaken on the acceptability of self-sampling in specific ethnic minority groups in the UK, as cervical screening 'never attenders' from these communities may be less likely to choose self-sampling than white women [23], and the reasons around this inequity need to be explored. This is particularly important in the context of

declining UK screening numbers, as women from ethnic minority groups in the UK have been associated with overall lower screening attendance [29, 30], with fear of pain, embarrassment and need for a female healthcare professional noted as barriers, to which hrHPV self-sampling may provide an alternative option to bridge these gaps. Our research group is undertaking a follow-on in-depth qualitative research study of such barriers: 'Understanding knowledge, beliefs, values and barriers towards cervical cancer screening, self-sampling and HPV vaccination amongst migrant Muslims and stakeholders in south-west London: An in-depth qualitative interview study' (IRAS reference 306176).

Furthermore, with regards to the impact in low-and-middle-income-countries where the proportion of screening non-attenders is significantly higher [2, 22], self-sampling may offer a highly acceptable alternative choice, that does not require a healthcare professional, and may therefore better serve hard to reach communities that face barriers to screening access and HPV vaccination.

There are a number of limitations in this analysis: Lack of data collection on demographics of the respondents was a significant limitation and would have provided increased depth of understanding of the contributory factors to women's screening preference, and there has been work on this in a number of previous studies [14, 20–23, 28]. These studies note a likely heterogeneity in results depending on demographics including social background, age, number of partners, and menopausal status for example, and there is a notable lack of acceptability data in participants from more deprived, non-English speaking and ethnic minority groups, which we are attempting to address though the in-depth qualitative study noted above.

Overall, this was a relatively small sample size of 207 respondents, with 195 (94%) completing the questionnaire in full. Strengths of this study however, are that this represents a good proportion of those recruited to the larger cervical cancer screening study, and the acceptability data showed good internal reliability, but larger sample sizes are needed to give a more accurate reflection of the screening-eligible population which is estimated to be around 5 million women and people with a cervix [3]. It is also noted that the participant cohort may be biased as they had attended initial screening and now colposcopic follow-up, as discussed earlier, though this does provide a unique insight into a population from a diverse area of London, who are mostly referred with cervical cell changes on pap smear, and may need increased surveillance.

## Conclusion

In conclusion, this study shows a highly acceptable method of hrHPV screening in the colposcopy clinic attender population but also the need for information about test accuracy to reduce a major barrier to self-sampling. This method has the potential to ease the burden on the NHS Cervical Screening Programme (NHSCSP) two-fold: Firstly, by reducing the number of face-to-face contacts required for cervical screening (and the need for clinician taken samples); and secondly resulting in a reduced cost of screening [31]. Above all, self-sampling could provide more equitable access to screening in under-screened populations. Further research is needed to understand patient screening preferences and interventions to address cervical health literacy across a diverse population. Any shift in the cervical screening paradigm in the UK requires comprehensive education and support for patients, the general public and healthcare providers.

## Supporting information

**S1 Questionnaire. Low vaginal self-sampling acceptability questionnaire St George's Hospital Foundation NHS Trust.**
(DOCX)

**S1 Checklist. Inclusivity in global research.**
(DOCX)

## Acknowledgments

We would like to thank the study participants, and the patients and experts involved in the focus group contributing to the design of the questionnaire, specifically Liberty Lean, Julie Faerber, Janka Briestenska (Colposcopist), and Uzma Razat (from the St. George's Hospital Charity). We are grateful for the contribution of the colposcopy clinic staff who were always accommodating and motivated to allow time for discussion and informed consent. Finally, the advice and knowledge resource sharing by Program ROSE team at University Malaya, Malaysia was invaluable in the design and implementation of this study.

## Author Contributions

**Conceptualization:** Sophie Webb, Nafeesa Mat Ali, David J. Clark, Megan A. Brown, Yolanda Augustin, Henry M. Staines, Sanjeev Krishna, Kevin Hayes.

**Data curation:** Sophie Webb, Nafeesa Mat Ali, Amy Sawyer.

**Formal analysis:** Sophie Webb, Nafeesa Mat Ali, David J. Clark.

**Funding acquisition:** Yolanda Augustin, Henry M. Staines, Sanjeev Krishna, Kevin Hayes.

**Investigation:** Sophie Webb, Nafeesa Mat Ali, Amy Sawyer.

**Methodology:** Sophie Webb, Nafeesa Mat Ali, Yin Ling Woo, Su Pei Khoo.

**Project administration:** Sophie Webb.

**Resources:** Yin Ling Woo, Su Pei Khoo, Kevin Hayes.

**Supervision:** Yolanda Augustin, Sally Hargreaves, Henry M. Staines, Sanjeev Krishna, Kevin Hayes.

**Validation:** Sophie Webb, Yolanda Augustin, Henry M. Staines, Sanjeev Krishna, Kevin Hayes.

**Visualization:** Sophie Webb, Yolanda Augustin, Sally Hargreaves, Sanjeev Krishna, Kevin Hayes.

**Writing – original draft:** Sophie Webb.

**Writing – review & editing:** Sophie Webb, Nafeesa Mat Ali, David J. Clark, Yolanda Augustin, Yin Ling Woo, Su Pei Khoo, Sally Hargreaves, Henry M. Staines, Sanjeev Krishna, Kevin Hayes.

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
