## [Decision Letter · Decision Letter 0]

19 Jan 2024

PGPH-D-23-02495

Patient preference and acceptability of self-sampling for cervical screening in colposcopy clinic attenders: a cross-sectional semi-structured survey

Dear Dr. Webb,

Thank you for submitting your manuscript to PLOS Global Public Health. After careful consideration, we feel that it has merit but does not fully meet PLOS Global Public Health’s publication criteria as it currently stands. Therefore, we invite you to submit a revised version of the manuscript that addresses the points raised during the review process.

Dear Author,

I have reviewed the manuscript submitted for the consideration for publication entitled''Patient preference and acceptability of self-sampling for cervical screening in colposcopy clinic attenders: a cross-sectional semi-structured survey''.

The reviewer(s) suggest some minor revisions to your manuscript. Therefore, I invite you to respond to the reviewer(s)' comments and revise your manuscript.

Please ensure that your decision is justified on PLOS Global Public Health’s publication criteria and not, for example, on novelty or perceived impact.

We look forward to receiving your revised manuscript.

Kind regards,

Esra Keles

Academic Editor

Journal Requirements:

2. We do not publish any copyright or trademark symbols that usually accompany proprietary names, eg  ©, ®, ™  (e.g. next to drug or reagent names). Please remove all instances of trademark/copyright symbols throughout the text, including ® on page 7.

Reviewers' comments:

Reviewer's Responses to Questions

**Comments to the Author**

1. Does this manuscript meet PLOS Global Public Health’s publication criteria? Is the manuscript technically sound, and do the data support the conclusions? The manuscript must describe methodologically and ethically rigorous research with conclusions that are appropriately drawn based on the data presented.

Reviewer #1: Yes

Reviewer #2: Yes

Reviewer #3: Yes

2. Has the statistical analysis been performed appropriately and rigorously?

Reviewer #1: Yes

Reviewer #2: Yes

Reviewer #3: Yes

3. Have the authors made all data underlying the findings in their manuscript fully available (please refer to the Data Availability Statement at the start of the manuscript PDF file)?

Reviewer #1: Yes

Reviewer #2: Yes

Reviewer #3: Yes

4. Is the manuscript presented in an intelligible fashion and written in standard English?

Reviewer #1: Yes

Reviewer #2: Yes

Reviewer #3: Yes

5. Review Comments to the Author

Reviewer #1: Patient preference and acceptability of self-sampling for cervical screening in colposcopy clinic attenders: a cross-sectional semi-structured survey

Line 36-40: The authors should first start the paragraph by explaining the variations in options of self-sampling available to the participants before describing the proportion who accepted what type of sampling. This will allow clarity in the study and make it easier to understand.

Line 44: What does the author mean by “the lack of strong preference for screening method…” do you mean those who did not have any preference?

Line 61: This paragraph reviewing the decline in the screening rate in the recent years should also discuss aside COVID-19 what are the other possible factors that could lead to the decline in the screening rate, this can further buttress the need for the self-sampling technique and justify the need for this study. This was explained briefly in line 75 to 77 however, it was not detailed.

Method section, Measures; the available choices of sampling for the participants should be described when explaining the Questionnaire.

I would also suggest that a future recommendation should include experimental studies of training woman on self-sampling and accessing their level of confidence before and after the training. Evidence of this is in the result section, 53% didn’t have confidence and also the 43% Preference for clinician to take needs to still be explored, why the increased number of preference for a clinician.

This study is a very relevant one and will bring about a shift in the healthcare sector, which is relevant for cervical screening, but I believe the decline in the self-sampling preference will have a lot to do with self-confidence of performing the procedure.

Great work!

Reviewer #2: I have three main comments.

Line 94: This needs to be clarified. Are there two locations with 59% and 52% non-white British ethnicity?

Line 170: This is an important finding, especially given the racial diversity of the sample and the relatively high levels of racism in healthcare in Britain.

Line 262: I wish a deeper analysis on race in health in Britain could have been examined here, even if 2-3 sentences. There is a wealth of literature on this.

Reviewer #3: This is an important study on Patient preference and acceptability of self-sampling for cervical screening in colposcopy clinic attenders: a cross-sectional semi-structured survey

The study aims to seek views of the end users on a new point of care.

The manuscript is well written and has coherence

The methods and results are well presented

The conclusion is in keeping with the aim of the study

The authors may clarify in the discussion

How was your study different from the one from Malaysia from which your methods were adopted

What are the strength of this study?

6. PLOS authors have the option to publish the peer review history of their article (what does this mean?). If published, this will include your full peer review and any attached files.

**Do you want your identity to be public for this peer review?** For information about this choice, including consent withdrawal, please see our Privacy Policy.

Reviewer #1: **Yes: **Queen Esther Adeyemo

Reviewer #2: No

Reviewer #3: **Yes: **ESTER LILIAN ACEN

---

## [Editor Report · Decision Letter 1]

24 Apr 2024

Patient preference and acceptability of self-sampling for cervical screening in colposcopy clinic attenders: A cross-sectional semi-structured survey

PGPH-D-23-02495R1

Dear Miss Webb,

We are pleased to inform you that your manuscript 'Patient preference and acceptability of self-sampling for cervical screening in colposcopy clinic attenders: A cross-sectional semi-structured survey' has been provisionally accepted for publication in PLOS Global Public Health.

Best regards,

Esra Keles

Academic Editor

Dear Authors,

The required revisions have been made, and your article is suitable for publication.

Sincerely